# Utilizing Electrochemical-Based Sensing Approaches for the Detection of SARS-CoV-2 in Clinical Samples: A Review

**DOI:** 10.3390/bios12070473

**Published:** 2022-06-29

**Authors:** Nor Syafirah Zambry, Godwin Attah Obande, Muhammad Fazli Khalid, Yazmin Bustami, Hairul Hisham Hamzah, Mohd Syafiq Awang, Ismail Aziah, Asrulnizam Abd Manaf

**Affiliations:** 1Institute for Research in Molecular Medicine (INFORMM), Health Campus, Universiti Sains Malaysia, Kubang Kerian 16150, Kelantan, Malaysia; norsyafirah@usm.my (N.S.Z.); fazlikhalid@usm.my (M.F.K.); 2Department of Medical Microbiology and Parasitology, Universiti Sains Malaysia, Kubang Kerian 16150, Kelantan, Malaysia; obandega@student.usm.my; 3Department of Microbiology, Faculty of Science, Federal University of Lafia, Lafia PMB 146, Nasarawa State, Nigeria; 4School of Biological Sciences, Universiti Sains Malaysia, Gelugor 11800, Pulau Pinang, Malaysia; ybustami@usm.my; 5School of Chemical Sciences, Universiti Sains Malaysia, Gelugor 11800, Pulau Pinang, Malaysia; hishamhamzah@usm.my; 6Collaborative Microelectronic Design Excellence Centre (CEDEC), Sains@USM, Universiti Sains Malaysia, Bayan Lepas 11900, Pulau Pinang, Malaysia; mohdsyafiqawang92@student.usm.my

**Keywords:** COVID-19, SARS-CoV-2, diagnostic methods, electrochemical biosensor, point of care (POC), miniaturised electrochemical sensor, microfluidic electrochemical devices

## Abstract

The development of precise and efficient diagnostic tools enables early treatment and proper isolation of infected individuals, hence limiting the spread of coronavirus disease 2019 (COVID-19). The standard diagnostic tests used by healthcare workers to diagnose severe acute respiratory syndrome coronavirus-2 (SARS-CoV-2) infection have some limitations, including longer detection time, the need for qualified individuals, and the use of sophisticated bench-top equipment, which limit their use for rapid SARS-CoV-2 assessment. Advances in sensor technology have renewed the interest in electrochemical biosensors miniaturization, which provide improved diagnostic qualities such as rapid response, simplicity of operation, portability, and readiness for on-site screening of infection. This review gives a condensed overview of the current electrochemical sensing platform strategies for SARS-CoV-2 detection in clinical samples. The fundamentals of fabricating electrochemical biosensors, such as the chosen electrode materials, electrochemical transducing techniques, and sensitive biorecognition molecules, are thoroughly discussed in this paper. Furthermore, we summarised electrochemical biosensors detection strategies and their analytical performance on diverse clinical samples, including saliva, blood, and nasopharyngeal swab. Finally, we address the employment of miniaturized electrochemical biosensors integrated with microfluidic technology in viral electrochemical biosensors, emphasizing its potential for on-site diagnostics applications.

## 1. Introduction

The novel coronavirus SARS-CoV-2 that caused COVID-19 disease was firstly discovered in Wuhan, Hubei Province, China, in December 2019. The World Health Organization (WHO) declared COVID-19 to be a pandemic due the capability of SARS-CoV-2 viral to rapidly spread worldwide [1]. The virus can be transmitted through respiratory droplets during coughing, talking, and sneezing, with the incubation time from 2 to 14 days [2,3]. People infected with this virus may present with very mild clinical symptoms, e.g., flu, headache, sore throat, cough, fever, and diarrhea, to life-threatening conditions such as multi-organ dysfunction and interstitial pneumonia that is possibly caused a thrombophilic vasculitis in the lung [4,5,6,7]. In most cases, SARS-CoV-2 patients demonstrate acute respiratory distress syndrome (ARDS). In this condition, individuals find it difficult to breathe since the oxygen level in blood keeps decreasing due to consumption by the virus during their replication [8]. The mortality rate of SARS-CoV-2 was varied depending on the geographic area [9]. Since their emergence, more than 20 million people worldwide have been infected and approximately more than 800,000 deaths were recorded [1]. The numbers kept elevated until presently, as no specific antiviral treatment is available for this virus [10]. Accordingly, the COVID-19 pandemic threatened the world by badly impacting the public health sector, which subsequently affected socio-economic aspects and national financial policies [11].

To overcome the current global pandemic, various diagnostic tests are developed to give extremely fast and accurate detection. The effective diagnostic systems enable the immediate isolation of individuals that present mild infection symptoms through a strict quarantine and thus interrupt the transmission chain of COVID-19 to the surrounding community [12,13]. Despite a high rate of disease spreading, the main challenge for the COVID-19 diagnostics lies in the asymptomatic individuals with SARS-CoV-2 infections. The asymptomatic individuals have higher chances to spread the virus efficiently, hence causing a predicament to control the outbreak of the disease [14]. Therefore, early diagnostic tests with high specificity and sensitivity, precise, and rapid are crucial for mass screening of SARS-CoV-2, to identify positive cases which enable contact tracing and containment. Such situations can curb the spreading and infection rate of the virus and thus provide ample time for developing vaccines or treatments to control this contagious virus [15,16]. Presently, numerous diagnostic tests are available for the early detection of virus infection. The diagnostic tests for SARS-CoV-2 mostly relied on detecting viral nucleic acid (DNA or RNA) and antigens or antibodies produced upon exposure to infection [17,18]. To date, healthcare workers have extensively used the quantitative real-time reverse transcription-polymerase chain reaction (real-time RT-qPCR), and enzyme-linked immunosorbent-assay (ELISA)-based testing to diagnose SARS-CoV-2 [19,20,21]. Although these methods provide high sensitivities and reliable results, they are not preferable for rapid on-site diagnosis. This is due to some restrictions such as tedious sample preparation, long detection process, the requirement of well-trained staff, and sophisticated instruments [18,22]. Therefore, the biosensor particularly an electrochemical biosensor is seen as a good alternative to the existing diagnostic tests since it quickly diagnoses viral diseases with high selectivity and sensitivity [23,24]. 

A biosensor is an analytical electronic device that composed of three associated elements: a biorecognition molecule; a transducer (an electronic part that transfers a biochemical signal from the interaction between analyte and biorecognition molecule into an electronic signal); and a processor (amplifies and shows the analytical response signal that can be quantifiable) [25,26]. Compared to the existing COVID-19 detection methods, biosensors enable selective and sensitive detection of a targeted analyte cost-effectively and rapidly. Biosensors can perform either semi-quantitative or quantitative real-time analyses of analytes without the need for sample preparation and reagents. More importantly, they have the potential to enable in situ analyses, which are crucial features for point-of-care (POC) diagnostic [27,28,29]. Aside from medical and POC applications, these innovative bioelectronic devices have been extensively used in food processing, food safety, environmental monitoring, drug recovery, forensics, and biomedical research [25,30,31]. There are various types of biosensors, and it is generally categorized based on the type of biorecognition molecules immobilized, and transducer used [32]. Biorecognition molecules used in the biosensors consist of antibodies, nucleic acids, enzymes, biomimetic materials, and whole cells. The signal recognition of biosensors is achieved through various types of transducers including electrochemical, piezoelectric, optical, and thermometric [25]. The electrochemical transducer is the most popular biosensor, and they are extensively used to detect high numbers of biomarkers and infectious diseases. The implementation of electrochemical biosensors for the quantitative or semi-quantitative analysis of respiratory viruses had been proven due to their intrinsic strength such as simplicity in design, flexibility, miniaturized instrumentation, fast detection time, low cost, and high sensitivity [33,34,35]. 

This review focuses on the use of electrochemical biosensors for the ultrasensitive detection of SARS-CoV-2. Here, we collated recent research papers that described numerous electrochemical biosensor designs for early detection of this infectious virus in a variety of clinical samples (serum, saliva, urine, and nasopharyngeal swab). Factors that determine the ideal fabrication of SARS-CoV-2 electrochemical biosensors such as the working electrode materials, biorecognition elements, and transducing techniques will be discussed. The addition of nanostructured materials (graphene derivates, carbon nanotubes, gold nanoparticles, etc.) as the surface modifier on the miniaturized electrochemical biosensor enhanced the sensitivity of the electrochemical biosensor. Further integration of the miniaturized electrochemical platform with microfluidic technology and smartphone enables rapid and on-site detection of SARS-CoV-2.

## 2. SARS-CoV-2 Diagnostic Tests Advantages and Challenges

Table 1 summarises the pros and cons of the existing diagnostic tests and electrochemical biosensors for SARS-CoV-2 detection. Formerly, a computerized tomography (CT) scan has been remarkably used as a reliable diagnostic tool for screening individuals infected with COVID-19. CT scan provides information on the various organs cross-sectional images, included blood vessels, soft tissues, bones, and inside the body. Such images can give detailed information on pathophysiology to diagnose and evaluate the severity of disease [36]. Chest CT scan is known as the primary diagnostic test for COVID-19 [37]. Chest CT scan images of a COVID-19 patient usually show bilateral, peripheral, patchy consolidation, and basal predominant ground-glass opacities (GGOs) with the sub-pleural distribution [38]. The continuous observation of chest CT scans could facilitate fast diagnosis, monitor disease progress, and determine the suitable treatment for COVID-19 patients. Nevertheless, the chest CT scan images cannot distinguish the SARS-CoV-2 from other acute respiratory diseases and often detect the patients at an advanced stage of infection [39]. Furthermore, the imaging features of a SARS-CoV-2 patient’s chest may vary according to the viral concentration and severity of infection after the appearance of severe symptoms. For example, during the early stage of infection (0–4 days), the CT scan images shows “halo” sign in the right upperlobe (25%). The ground-glass opacities, reticular pattern, crazy-paving pattern, subpleural lines, pleural thickening and fibrosis were commonly observed during the intermediate/progressive (5–9 days) and advanced (≥10 days) stages of the SARS-CoV-2 infection [40]. Other drawbacks of this method are the requirement of advanced, skilled personnel and expensive equipment [41]. Therefore, this method cannot be implemented in small rural towns that have limited technology.

The current standard diagnostic test for detecting SARS-CoV-2 depends on two strategies: detection of viral antigens and nucleic acid. Another method is the detection of specific antibodies produced against the virus by the patient’s immune response [17,42]. The real-time reverse transcription-polymerase chain reaction (real-time RT-PCR) is known as a standard gold method for diagnosing SARS-CoV-2 infection [43]. The nucleic acid-based method was acknowledged to rapidly detect the SARS-CoV using multiple primers and probe sets in distinct regions of the SARS-CoV genome. The current real-time RT-PCR assay has been established to detect the structural proteins and accessory genes (spike (S) protein, nucleocapsid (N), RNA-dependent RNA polymerase (RdRP), envelope (E), orORF1b, ORF8 genes), as these are specific biomarkers for SARS-CoV-2 [41,44]. The high selectivity of this technique distinguishes SARS-CoV-2 from other human and animal coronaviruses. To date, numerous quantitative RT-PCR (qRT-PCR) test kits are available in the market such as CDC 2019-Novel Coronavirus Real-Time RT-PCR Diagnostic Panel (Centers for Disease Control and Prevention, Atlanta, GA, USA), ExProbeTM SARS-CoV-2 Testing Kit (TBG Diagnostics Corp., New Taipei City, Taiwan), Xpert Xpress SARS-CoV-2 test (Cepheid, Sunnyvale, CA, USA), Abbott RealTime SARS-CoV-2 RT-PCR Kit (Abbott, Chicago, IL, USA), TaqPath COVID-19 Combo Kit (Thermo Fisher Scientific, Life Technologies Corporation, Carlsbad, CA, USA) and PerkinElmer^®^ New Coronavirus Nucleic Acid Detection Kit (PerkinElmer, Inc., Austin, TX, USA) [43]. Despite these methods’ high selectivity and sensitivity, the assay requires a higher concentration of target molecules, lengthy processing time, sophisticated, expensive laboratory equipment, and skilled personnel [45,46]. To overcome these issues, isothermal amplification techniques such as nucleic acid sequence-based amplification (NASBA), loop-mediated isothermal amplification (LAMP), strand displacement amplification (SDA), nicking enzyme amplification reaction (NEAR), rolling circle amplification (RCA), and helicase dependent amplification (HDA) are developed for detection of SARS-CoV-2 infection [47]. Although the detection methods are highly sensitive, more specific, and reaction efficient, the current technology still faces some limitations [48,49]. For example, the ID NOW® instrument can provide a rapid detection time (within 13 min) [50]. However, the time taken, and labour needed for sample preparation, reagent loading, and the instrument’s set-up does not count. [47,51,52]. In addition, the technology used is costly and less attractive for mass screening applications.

**Table 1 biosensors-12-00473-t001:** Summary of advantages and drawbacks of the existing diagnostic methods and electrochemical biosensors for SARS-CoV-2 detection.

Detection Method	Target	Laboratory or Point-of-Care (POC)	Quantitative	Advantages	Cost of Testing	Drawbacks
CT scan	Chest	Laboratory	No	High sensitivity	High	Lack of specificity Require sophisticated and expensive machines Need trained personnel to interpret the results Exposed to the radiation
RT-PCR	Nucleic acid	Laboratory	Semi-quantitative	Highly specific and sensitive Suitable for early infection Can detect the viral particles that cannot be cultured by cell culture method	High	Require sample preparation and purification Need specific reagents Require sophisticated and expensive machines Need skilled personnel Chances of false results are higher for mixed infection cases Longer analysis time (~50 min to 4 h) Not suitable for mass population Not suitable for large scale screening for multiple samples
ELISA	Antigen Antibody	Laboratory	Semi-quantitative	Suitable for monitoring the immune response Suitable for sero-surveillance	High	Require sample preparation and purification Low specificity High risk of cross-reactivity Longer analysis time (~2 to 5 h) Not suitable for large scale screening for multiple samples
Electrochemical biosensor	Any analyte depending on the biorecognition element	POC	Yes	Rapid response time (~10 s to 1 h) Highly specific No need complex reagents and sample preparation Miniaturization capability	Low	Sample matrixes affect the sensitivity of assay Low stability

CT: Computerized tomography, RT-PCR: Real-time polymerase chain reaction, ELISA: Enzyme-linked immunosorbent assay. Adapted from reference [43,53,54,55,56,57].

The most common antibody test used to diagnose SARS-CoV-2 infection is direct enzyme-linked immunosorbent assay (ELISA) [58]. In direct ELISA, the specific antigen (protein) is immobilised on a solid matrix (ELISA plates), then incubated with the primary antibody and followed by the addition of the substrates that can generate a color (e.g., horseradish peroxidase and alkaline phosphatase). The changes in color in the ELISA plates indicate the binding of a specific antigen to the antibody due to the enzymatic cleavage of a chromogenic substrate (Figure 1) [59,60]. ELISA provides an accurate result for infected patients and allows high throughput SARS-CoV-2 screening as it could process many samples simultaneously. The rapidness and sensitivity of this test allow the ELISA kits to be commercially available from various manufactures, namely Thermo Fisher Scientific, Abcam, and Merck [61]. Nonetheless, the assay is restricted to a single antigen per well and requires the complex production of antibodies [62,63]. For an asymptomatic person, ELISA might show an inaccurate result because the antibody only shows response after the 10th day of infections. Moreover, this assay might have a possibility of false-positive results due to the interferences with other proteins in serum samples of SARS-CoV-2 and other coronaviruses [42,64]. 

The limitations and problems related to the above-mentioned diagnostic tests encourage researchers nowadays to develop more robust and accurate detection systems for efficiently screening COVID-19 patients. The robust screening system enables effective isolation and prompt treatment to the patient, which subsequently breaks the COVID-19 transmission chain. To achieve the goal, the designed diagnostic assay must possess the following characteristics; the affordable sensing technologies (minimal cost and fewer demands for personnel and instrument), high sensitivity (ability to recognize infected individual, true positive rate), high specificity (ability to recognize non-infected individual, true negative rate), equipment-free (portable device), the faster response time (ideally in a minute or maximum is an hour), and deliverable (user-friendly). These characteristics are known as ASSURED criteria [47,65,66,67]. Based on these requirements, electrochemical biosensors are seen as suitable candidate with favorable characteristics.

The development of electrochemical biosensors has aroused the attention of most researchers nowadays due to their remarkable advantages such as high analytical sensitivity and selectivity, the simplicity in design, low cost of the device, miniaturization capacity, and inherent sustainability, due to the use of low reagents, and sample volumes, both in its development and application [68,69,70]. The features of electrochemical biosensors satisfy the industry requirement that emphasizes the rapidness of the method and the initial investment cost, technical support, and ease of handling [42,71,72]. Due to their significant properties, electrochemical biosensors have been established as a robust diagnostic device to test numerous clinical samples to detect infectious diseases caused by pathogenic viruses and bacteria [73]. Notably, the electrochemical biosensor has been emerging as an alternative tool for monitoring SARS-CoV-2 infection [74].

Despite having superior diagnostic features, the electrochemical biosensors platform faced several challenges such as the sample preparation procedure, stability and selectivity were dependent on the biorecognition molecules used. Moreover, the contamination of these biorecognition molecules might cause the performance of electrochemical biosensors to not be as effective [70]. Electrochemical viral infectious disease diagnosis also encountered difficulty in detecting pathogen directly from raw samples due to the presence of interferences such as vitamins and proteins that are commonly present in the bio-fluids. Thus, several separation steps are needed before the detection process, which subsequently prolongs the detection time [75]. The integration of the latest microfluidic technology with an electrochemical sensing platform could overcome this limitation as it facilitates the method for sample separation and electrochemical detection in one device [76]. Besides the biorecognition molecules, the electrochemical biosensor can achieve high selectivity and sensitivity due to easy modification of the electrode surface with nanomaterials in the lab scales [70]. However, the mass manufacturability aspect should be considered to produce a robust diagnostic tool for the end-users. The following section will discuss the ideal SARS-CoV-2 electrochemical biosensors system using the most robust electrode materials and specific biorecognition molecules that affect the sensitivity and selectivity of electrochemical detection.

## 3. SARS-CoV-2 Electrochemical Biosensors

Electrochemical biosensors are known as the most promising and highly sensitive transduction systems for the early detection of SARS-CoV-2. This is due to its capability in detecting the target analyte at a very low concentration with low power consumption [77]. Generally, electrochemical biosensors are biosensing devices that transform the biochemical reactions between biorecognition molecule and target analyte into measurable signals via a current, voltage, or charge transfer resistance [78] (Figure 2). The generating signals are proportional to the concentration of target analytes in the biochemical reaction [28,79]. The utilization of electrochemical biosensors to detect a different group of viruses has been extensively reported in the literature. For instance, Anusha et al. [80] discussed numerous designs of electrochemical biosensing platforms and the function of biorecognition molecules in detecting the dengue virus. The recent advancement in developing robust electrochemical sensors strategies to detect the Zika virus has been extensively reviewed by Kaushik et al. [81]. Moreover, Rasouli et al. [77] highlight the progress of electrochemical DNA biosensors in monitoring the human papillomavirus virus. Due to its excellent performance in pathogenic virus detection, electrochemical biosensors are suggested as one of the promising diagnostic tests for the real-time observation of SARS-CoV-2. Many pieces of literature reported on the potential implementation of electrochemical biosensors for the detection of SARS-CoV-2. For example, Mahshid et al. [53] extensively reviewed the potential application of electrochemical biosensors for monitoring SARS-CoV-2. The most favourable electrochemical biosensors design was thoroughly discussed by Kotru et al. [82]. Indeed, Khan et al. [79] and Imran et al. [24] described the performance of different types of electrochemical biosensors for SARS-CoV-2 and other viral pathogens such as Ebola, influenza, Zika, and HIV.

In an electrochemical biosensors system, the target analyte can be detected either through a labeled system (indirect sensing) or a label-free system (direct sensing). The label-based electrochemical biosensor was constructed by adding a second probe or specific label (e.g., ferrocene, enzyme, metal nanoparticles, etc.) to the biorecognition element specific to the target analyte. In this sensing system, the concentration of the target analyte was indirectly measured based on generating signals from a specific label, and PBS was mostly used as an electrolyte solution for electrochemical measurement. Formerly, most of the highly sensitive electrochemical biosensors were fabricated based on a labelled system due to the potential for signal amplification and high resistance to non-specific binding as the detection of the analyte depends on two independent binding events. Nonetheless, the fabrication of this type of electrochemical biosensor is expensive because it requires two biorecognition systems, and multiple steps are needed during the surface functionalization of the working electrode. Furthermore, the additional preparation steps for complex labelling might prolong the time needed to construct the electrochemical biosensors. Moreover, electrochemical detection can only be carried out by highly trained personnel to ensure the quality of the analysis and interpretation of the results [83,84]. Hence, they are not preferable for real-time measurements. 

Unlike label-based electrochemical biosensors, a label-free sensing system can minimize the preparation time and cost of analysis due to the elimination of complex labels and the requirement of only one biorecognition molecule. Such a sensing system can quantify the target analyte directly based on the electrical signals produced during the biochemical reaction without any labelling and chemical modification. The electrochemical detection in a label-free sensing system directly relies on the catalytic transfer of electrons between the surface of the working electrode and the active surface of a biorecognition molecule. The measurement of this sensing system often employs ferro/ferricyanide as a redox probe [22,85,86]. The capability of label-free electrochemical biosensors for rapid detection of target analyte enables them possible for real-time measurements and as a POC device [83,87]. Due to their significant advantages, label-free electrochemical biosensors have become a popular choice for electrochemical-based diagnostic of SARS-CoV-2 infection.

The fabrication of label-free electrochemical biosensors critically relies on several components such as the transducer (working electrode), biorecognition molecules, and electrochemical transducing techniques. These sensing platforms ensured more sensitive detection by fabricating the working electrode with materials that possess excellent electronic performance and applying the high specificity and affinity of biorecognition elements [88]. Table 2 demonstrates numerous types of electrochemical biosensors for detecting SARS-CoV-2 in clinical samples such as saliva, nasopharyngeal swab, serum, throat swab, urine, and feces. The limit of detections (LOD) was in the range from nanomolar to femtomolar using various fabrication strategies and electrochemical transducing techniques that will be discussed in the following subsections.

### 3.1. Transducer (Working Electrode) and Electrochemical Transducing Methods

The transducer or working electrode can be fabricated with semiconducting and conducting materials ranging from metals (e.g., platinum and gold) to non-metals (e.g., carbon) using diverse sizes of materials (from bulk to micro and nanostructured materials) [67,70,88]. Additionally, polymer electrodes and ceramic electrodes are often chosen in the fabrication of working electrodes of electrochemical biosensors due to their advantageous properties such as stability, biocompatibility, and tuneable electric conductivity [120]. The selection of working electrode materials will determine the performance of the electrochemical biosensor including the rate of heterogeneous electron transfer (affect the sensitivity and detection time), double-layer capacitance (affect the limit of detection), the character of the coupling chemistry required to attach the biorecognition molecules, and the propensity towards non-specific binding [67,88]. 

In electrochemical biosensors, the changes in electrical properties of the working electrode can be measured based on several transducing methods such as voltammetry (measure the current at a different range of potential), amperometric (measure the current at constant potential), potentiometric (measure the charge potential at fixed current), and conductometric (measure the conductivity of working electrode at varying of frequencies) depending on the detection principle and application of electrochemical biosensors. Among all the transducing techniques, the voltammetric techniques, namely cyclic voltammetry (CV), square wave voltammetry (SWV), and different pulse voltammetry (DPV), are often chosen in electrochemical diagnostics sensing due to the simplicity of the method and low-cost instrumentation (only a potentiostat is needed) [27,121,122,123]. Therefore, the detection times could be completed in a short period, and detection costs can significantly be reduced [124]. Moreover, these methods are significantly used in the fabrication of label-free electrochemical biosensors for viral detection [85]. The mode of applied potential differentiates among the CV, DPV and SWV techniques. In CV, the sample sweeps through varying potentials, whereas the potential of sample pulses from one potential to another in both DPV and SWV techniques [125]. 

CV technique provides information on the preliminary redox characteristics of the biorecognition elements or target analyte. Cyclic voltammograms can demonstrate the electrochemical characteristics and reversibility of the redox reaction (semi or quasi reversible, irreversible, completely reversible). Moreover, the CV technique is often used to measure the surface area of various materials that are functionalized onto the electrodes to observe the improvement of the electrochemical performance [75,126]. CV is commonly employed in diagnostics approaches to characterise target chemical substances and assess the electrochemical mechanism of chemical reactions. Nevertheless, CV is not recommended for quantitative detection of target chemical substances as it possess low resolution and sensitivity [107]. DPV and SWV are favored voltammetric methods used in electrochemical diagnostics due to their high resolution and sensitivity [75,107]. In DPV, the peak voltammogram was obtained through the measurement of differences between two current responses recorded in each pulse period [127]. Meanwhile, SWV voltammograms were obtained by plotting the differences between two current responses recorded in each pulse period against the staircase potential. The capability of DPV and SWV to reduce the effect of capacitive current and increase the signal-to-noise ratio by attenuating background current makes these voltammetry techniques important over other electrochemical techniques [127,128]. Due to these characteristics, DPV and SWV enable the detection of SARS-CoV-2 up to picomolar or femtomolar concentration [75].

Besides voltammetric techniques, the impedance technique, namely, electrochemical impedance spectroscopy (EIS), has been widely used in electrochemical diagnostics. Generally, it measured the impedance or capacitance of the electrochemical biosensors system by imposing a sine wave with an amplitude range from 5 to 10 mV. Through EIS, the interfacial characteristics (adsorption or desorption) between the electrode and electrolyte could be determined. Moreover, the EIS data can give information on the kinetic or mechanistic aspects and the electrochemical redox reaction rates [129]. Similar to CV, EIS measurement was also performed throughout every step of the electrochemical biosensor fabrication to follow the electrode surface modification. For example, Abrego-Martinez et al. [106] have performed the CV and EIS analyses in an electrolyte solution containing 5 mM [Fe(CN)_6_]^3−/4−^ to evaluate the success of each fabrication step of aptasensor for monitoring the SARS-CoV-2 pseudovirus. The developed aptasensor could detect the spike protein of SARS-CoV-2 within 40 min of incubation time with a LOD of 66 pg/mL by EIS measurement. 

### 3.2. Biorecognition Molecules Used for Fabrication of SARS-CoV-2 Electrochemical Biosensor

Until now, the immunoassay and nucleic-acid-based assay have been extensively utilized for clinical detection of SARS-CoV-2. Hence, the integration of these assays with an electrochemical sensing platform could overcome the existing limitations and drawbacks. This is due to the fact that electrochemical biosensors can provide more precise and sensitive results, as well as lower testing costs, user-friendly due to simple instrumentations, robust diagnosis, and rapid response time [26,130]. In the fabrication of electrochemical biosensors, DNA or RNA, aptamer, antibodies, and peptides are the common biorecognition elements used to detect various pathogens, including human coronaviruses [115,131]. Table 3 summarises the advantages and limitations of different biorecognition elements in the fabrication of electrochemical biosensors. Two types of electrochemical biosensors namely electrochemical immunosensor and electrochemical DNA sensors mainly utilized for SARS-CoV-2 detection are extensively discussed in the following section. 

#### 3.2.1. Electrochemical Immunosensor

Antibodies are recognized as the biorecognition molecules for electrochemical immunosensors. They are often chosen in most bioanalytical labs and commercial diagnostic kits to detect proteins, viruses and cancer cells because of their good affinity, high sensitivity, and specificity [132]. Among various types of antibodies, monoclonal antibodies (mAb), polyclonal antibodies (pAb) and antibody single-chain Fv (scFv) fragments are found as the most significant antibodies used to detect the respiratory virus infection. In particular, the reaction of mAb is highly specific compared with pAb as it can only bind with a single epitope. Thus, there are no chances of cross-reaction. Unlike mAb, pAb can recognize various epitopes on a single antigen [133]. Due to this characteristic, pAb is broadly applied in biosensors fabrication since its production cost is less expensive and can be mass-produced. The scFv fragments are preferably used for antigen capture compared with the whole antibody due to their smaller size and low variability [74,134]. A good affinity of antibodies towards antigens (can achieve nanomolar level) yielded numerous types of electrochemical biosensors. It can be in the form of enzyme-amplified ELISA to label-free formats [23]. Nevertheless, the antibody-based assay faced some limitations such as the production process being complex and expensive. Moreover, the addition of signals tags to the antibodies might disrupt their affinity [135]. 

In an electrochemical immunosensor, specific antibodies are immobilized onto the surface of the working electrode using several techniques such as direct absorption, magnetic beads (MBs) or self-assemble monolayer (SAM). They bind to specific target analytes (antigens) to generate electrical signals that can be measured using various transducing methods (Figure 3) [74,79]. Among the immobilization techniques of antibodies, SAMs of alkanethiols is mostly applied due to the capability to create strong covalent bonds on the surface of the working electrode in a simple way [136]. The generated signals in immunosensing assay are correlated to the reaction rate of antigen-antibody. Electrochemical immunosensors have been proven to detect most infectious diseases due to their excellent sensitivity, fast detection time, and the ability of miniaturization [132]. 

Accordingly, numerous immunosensors have been developed for the detection of various groups of viruses. The first immunosensor for detecting SARS-CoV had been developed by Ishikawa et al. [137] based on field-effect transistor (FET). In this study, the biochemical reaction between antibody and antigen was quantified based on the conductometric method. The virus antigen nucleocapsid (N) protein was used as a target analyte to fabricate this electrochemical biosensor. Antibody mimic proteins (AMPs) were used as an alternative for conventional antibodies due to their simple and low-cost production, smaller size (usually 2–5 nm, less than 10 kDa), and has stability in a broad range of pH and electrolyte concentrations. The surface of the working electrode was modified with an AMP capture agent namely fibronectin-based protein (Fn) to improve the selectivity of the designed sensor. The well-developed sensing platform could detect the N protein in 44 µM of bovine serum albumin (BSA) as low as 2 nM within 10 min. 

Recently, Zaccariotto et al. [97], an impedimetric immunosensor was developed by immobilizing antibodies on modified glassy carbon for SARS-CoV-2 detection. Figure 4 illustrates the fabrication steps of the electrochemical immunosensor. The successful fabrication of the electrochemical immunosensor was evaluated using CV and EIS techniques. The modification of glassy carbon electrodes with reduced graphene oxide presents a low-cost diagnostic technology. Hence, this strategy can directly be implemented to single printed carbon electrodes to make the developed immunosensor become a POC device in the future. In this study, the fabricated impedimetric immunosensor was successfully detected the SARS-CoV-2 spike protein with LOD of 150 ng/mL in the range of linear concentration from 0.16 to 40 µg/mL. Besides an impedimetric method, the developed diagnostic platform can use a voltammetric technique such as SWV to detect this virus. The developed electrochemical immunosensor can detect this pathogenic virus at a concentration as low as 2.40 ng/mL using the SWV technique. The proposed immunosensor is a practical diagnostic test for screening SARS-CoV-2 infection as it demonstrated a good performance towards detecting the virus in the saliva samples. Hence, the detection process for the developed sensor is much simpler and rapid compared to other antibody-based antigen diagnostic methods. 

In another study conducted by Liv [96], a novel electrochemical immunosensor was developed to detect the spike protein of the SARS-CoV-2 antigen in spiked saliva and oropharyngeal swab samples. This research group fabricated the sensor using a glassy carbon electrode modified with gold and capped with cysteamine and glutaraldehyde. The developed immunoassay platform employs the voltammetric method (CV and SWV) to detect the spike antibody in synthetic and real samples. The viral surface spike (S) protein was chosen as a biomarker for SARS-CoV-2 detection since it is a core transmembrane protein of the virus and highly immunogenic [138]. The fabricated electrochemical immunosensor can detect the target SARS-CoV-2 antigen protein with a LOD of 0.01 ag/mL in both synthetic media and clinical samples. This is the best reported LOD compared to another voltammetric immunoassay [102,113,118]. In addition, the developed electrochemical immunosensor demonstrated a simple sample preparation and shorter detection time (~35 min) compared to other established electrochemical methods in the literature [93,102,115]. The fabricated biosensor also showed high selectivity by the ability to distinguish the SARS-CoV-2 and MERS-CoV antigens. The accuracy of the developed immunosensor for detection of SARS-CoV-2 using saliva and oropharyngeal swab samples proved the successful fabrication of this immunosensor as most of the existing diagnostic tools used blood and serum samples that required a lengthy sample preparation process. However, further study is needed to integrate the ultrasensitive developed immunosensor into a ready-to-use commercial sensor and kit.

Accordingly, numerous electrochemical immunosensors have been developed for high sensitivity detection of different kinds of viruses. The first electrochemical immunosensor for the detection of SARS-CoV had been developed by Ishikawa et al., (2009) based on field-effect transistor (FET) [137]. Here, the change in conductance generated by the interaction of antibody-antigen can be measured and correlated to the concentration of the analyte. This FET-based electrochemical sensor used the virus antigen nucleocapsid (N) protein as a SARS biomarker. As an alternative for conventional antibodies, antibody mimic proteins (AMPs) were utilized as affinity binding agents due to their simple and low-cost production, smaller size (usually 2–5 nm, less than 10 kDa), and stability to a broad range of pH and electrolyte concentrations. The surface of working electrode was modified with a fibronection-based protein (Fn) as AMP capture agent to selectively bind the antigen N protein. To further improve the immobilization of the AMPs and transduced signal, the exposed gate region of the FET-based immunosensor was modified with In_2_O_3_ nanowires on a Si/SiO_2_ substrate. The well-developed sensing platform was able to detect the N protein at sub-nanomolar concentrations in a shorter time without the need for labelled reagents. 

To date, a great number of electrochemical immunosensors have been reported for the detection of SARS-CoV-2 infection with good selectivity and LOD as presented in Table 2. In early 2020, Seo et al., (2020) have developed FET-based immunosensor for the detection of SARS-CoV-2 in clinical samples [91]. The surface of FET sensor was immobilized with a SARS-CoV-2 spike antibody through 1-pyrenebutyric acid N-hydroxysuccinimide ester (PBASE), an efficient interface coupling agent used as a probe linker. In this study, the viral surface spike (S) protein was chosen as a biomarker for virus detection because it is a major transmembrane protein of the virus and highly immunogenic. Moreover, the S protein exhibits diverse amino acid sequences among coronaviruses and thus enabling the specific detection of SARS-CoV-2 [138]. This FET immunosensor can detect the target SARS-CoV-2 antigen protein with LOD of 1 fg/mL in phosphate-buffered saline. In addition, the developed sensor demonstrated high sensitivity and selectivity by the ability to distinguish the SARS-CoV-2 antigen protein from those of MERS-CoV. The successful fabrication of this immunosensor enabling sensitive detection of the SARS-CoV-2 virus in clinical samples with LOD of 2.42 × 10^2^ copies/mL without sample pre-treatment or any labelling.

#### 3.2.2. Electrochemical DNA Sensor

Viral nucleic acids are considered the most appropriate biorecognition elements for screening SARS-CoV-2 infection since the IgM and IgG immune responses are very low during the early stage of infection [23]. The common biorecognition elements that have been used in viral nucleic acid-based electrochemical sensors are ssDNA, RNA, peptide nucleic acid, and hairpin DNA [139]. Among them, ssDNA probes have been widely utilized in the construction of electrochemical-sensor-based nucleic acid which is known as electrochemical DNA sensor [74]. In an electrochemical DNA sensing, the ssDNA probe (15–30 kb) was immobilized onto the surface of the working electrode to recognize its complementary ssDNA target and produced a double-strand DNA (dsDNA) (Figure 5). The main interaction that occurred between the ssDNA probe and its complementary ssDNA target is called DNA hybridization. This hybridization reaction can be transferred into a quantifiable electrical signal via electrochemical techniques. The electrical signals were proportional to the concentration of viral nucleic acids [140,141]. Generally, the electrical signals are generated from the electron transfer of the redox-active probe with the electrode. In most electrochemical detection, [Fe(CN)_6_]^3−/4−^ and [Ru(NH_3_)_6_]^3+/2+^ complexes re used as redox-active probes [142]. The immobilization of ssDNA onto the surface of the working electrode is a crucial step in fabricating an electrochemical DNA sensor because it will determine the excellent reactivity and orientation of the DNA probe to hybridize with its ssDNA target [143,144]. In an electrochemical DNA sensor, the ssDNA can be immobilized onto the surface of the working electrode through several techniques such as adsorption, covalent-bonding, and avidin-biotin interaction [145]. Among the techniques, chemisorption is widely applied for the covalent immobilization of thiol-modified DNA probe onto the gold electrode surface to form a self-assembly monolayer (SAM) due to the high-affinity interaction between the thiol group and gold electrode [146,147].

In early 2005, Abad-Valle et al. [148] developed the electrochemical DNA sensor to detect the SARS-CoV. This work designed the surface of the working electrode using 100 nm sputtered gold films. The gold surface was immobilized with a labelled thiolated DNA probe and hybridized with 30-mer complementary ssDNA that encodes a short lysine-rich region as target molecules. The electrochemical detection was measured indirectly using an alkaline-phosphatase-labelled streptavidin that converts a substrate into an electroactive product. The fabricated electrochemical DNA sensor successfully detects the SARS virus-specific sequence within 1 h with a LOD of 6 pM. Among many types of electrochemical DNA sensors, aptamers stand as the ideal probe on the sensing surface due to their extraordinary characteristics including high affinity and specificity to their target molecules, resistance to a wide range of temperatures, and can be easily modified by chemical groups for immobilization or labelling purposes [149]. Indeed, aptamers can be designed for a wide variety of targets, including antibodies, protein, enzymes, amino acids, growth factors, cancer biomarkers, toxins, metal ions and low molecular weight vitamins [150,151,152,153,154,155,156]. Owing to these properties, they exhibit various recognition mechanisms during electrochemical detection. In most electrochemical detections, their sensing mechanism relies on target-induced conformational changes that bring a redox reporter close to an electrode surface, triggering an increase in electrochemical signal [157]. The aptamer can be either artificial synthesis in vitro or biochemically synthesis via the systematic evolution of ligands by exponential enrichment (SELEX) process [131]. 

Unlike antibodies, aptamer offers enormous advantages such as small size, short synthesis time, cost-saving production, not involving animal production, and no batch-to-batch variation. More importantly, they possess lower detection limits (can reach the zeptomolar (zM) level) which are necessary for developing the high sensitivity electrochemical biosensor [131,158]. The utilization of aptamer as a recognition element for virus detection has been well described and reported in several pieces of literature. For instance, Bhardwaj et al. [159] fabricated the electrochemical sensor using ssDNA aptamers that target recombinant influenza A mini-hemagglutinin (mini-HA) protein and whole H1N1 viruses for detection of influenza H1N1 viruses. In this study, the selected aptamer was immobilized onto the surface of the glass working electrode that was chemically modified with indium tin oxide (ITO). The fabricated aptasensor demonstrates the high sensitivity and selectivity detection of the H1N1 virus with the LOD of 3.7 plaque-forming units (PFU)/mL. Besides using a single DNA aptamer as a recognition molecule, the aptamer target-antibody sandwich technique is another common sensing mechanism applied. The utilization of duplex recognition patterns in electrochemical detection would enhance accuracy and selectivity and give the lowest LOD of electrochemical biosensors [74].

Idili et al. [95] developed an electrochemical aptamer-based (EAB) biosensor for the detection of SARS-CoV-2 spike (S) protein in undiluted biological fluids (serum and artificial saliva). In this study, a redox reported-modified aptamer (Atto MB2 (methylene blue derivative)-aptamer) was covalently attached onto the surface of the gold working electrode. The binding of spike protein will trigger a change in the aptamer conformation, subsequently bringing the redox reporter closer to the gold electrode surface and generating the electrochemical signals. The SWV analysis was used to characterize the performance of the EAB biosensor in detecting different concentrations of spike protein in PBS buffer to mimic the condition of real clinical samples. The developed EAB biosensor can detect the SARS-CoV-2 RBD target as fast as within seconds (15 s), demonstrating its ability for POC devices. Moreover, this sensing platform showed a similar analytical performance to other diagnostic methods that use antibody and aptamer as they can detect the SARS-CoV-2 antigen down to picomolar levels in serum, buffer, and 50% artificial saliva. 

Yousefi et al. [100] reported the first reagent-free electrochemical sensor using DNA aptamer-antibody conjugate as biorecognition elements to detect the SARS-CoV-2 infection (Figure 6). This research group successfully fabricated the electrochemical sensor that can directly detect the SARS-CoV-2 virus and its associated spike protein in processed and unprocessed patient saliva within 5 min, without any reagents. The designed sensor adopted a sensing mechanism that detects the potential triggered transport of a DNA-antibody conjugate. Here, an analyte-recognizing antibody was attached to a rigid, negative-charged linker composed of DNA and immobilized onto the surface of the gold working electrode. Ferrocene redox reporter molecule was attached to a thiolated DNA probe at 3′ to monitor the interaction on the surface working electrode. The electrochemical signals of the fabricated aptasensor were recorded using chronoamperometry (CA) using a potential window from 0 to +500 mV for 50 ms using Ag/AgCl as a reference electrode. Such electrochemical sensor demonstrates long-term stability as its performance was not much affected after 9 months of storage. Despite their benefits and versatility, aptamer also possesses some limitations that restrict their applications, such as being susceptible to nuclease degradation and not binding to the targets that lack functional groups [132].

## 4. The Advanced Electrochemical Sensing Technologies for Point-of-Care (POC) Detection of SARS-CoV-2

It is no doubt that the rise in SARS-CoV-2 outbreak cases demands rapid on-site and accurate diagnosis devices without the need for skilled technicians and sophisticated laboratories for high throughput sample screening [82,160]. In such situations, POC devices are urgently needed for the early detection of SARS-CoV-2 infection as this may be the best solution for now. Nowadays, the utilization of POC devices allows the diagnostic test to be performed near the patient site, providing faster results than conventional laboratory testing. Due to these features, they are significant in situations where rapid medical decisions need to be taken for example in emergency departments [161,162]. Notably, the use of POC diagnostic tools for early screening of viral infection in the population could potentially combat the spread of infection. Owing to the excellent diagnostic performance mentioned earlier, the electrochemical biosensors show potential and could be integrated into point-of-care testing (POCT) for SARS-CoV-2 diagnosis. The advanced design in electrochemical sensing platforms such as miniaturized sensors and microfluidic chips makes them a good alternative to the existing mainstream diagnostic COVID-19 methods [160]. As tabulated in Table 2, most of the developed SARS-CoV-2 electrochemical biosensors employed miniaturized electrode and microfluidic technologies for the ultrasensitive and on-site detection of this contagious virus that will be discussed further in the next section. 

### 4.1. Nanomaterials as the Surface Modifier on the Miniaturized Electrochemical Sensor

In the current pandemic crisis, most sensor research moves towards developing nano-enabled miniaturized electrochemical biosensors to gain more accurate and fast detection of the SARS-CoV-2. The introduction of nanotechnology greatly improves the performance of SARS-CoV-2 electrochemical biosensors by enabling the detection at a very low level (picomolar (pM) to femtomolar (fM) level) and high selectivity toward virus protein [89]. Nanomaterials possess unique physical and chemical surface characteristics compared with the bulk material such as diffusivity, solubility, optical, thermodynamic, colour and magnetic features [163,164]. In biochemical sensing, nanomaterials exhibit functional electrical and mechanical characteristics that are responsible for enhancing the electrochemical, optical, and magnetic properties of biosensors [165,166]. Particularly, the addition of nanomaterials onto the surface of the working electrode can enlarge the biocompatible areas with the target analytes (e.g., enzymes, proteins, antibodies, DNA, and cells) which subsequently enhanced the sensor reactivity and sensitivity in an electrochemical sensing platform [165]. Moreover, the implementation of nanotechnology in viral electrochemical biosensors can reduce the detection time and experimental cost since a small volume of samples is needed for analysis [165,167]. 

Various types of nanomaterials have been used for fabricating the SARS-CoV-2 electrochemical biosensors such as gold nanoparticles (AuNPs), quantum dots (QDs), carbon nanotubes (CNTs), and graphene or graphene-derived nanomaterials (Figure 7). In an electro- chemical sensing platform, graphene is a good surface modifier to interface with diverse biomolecules and cells [167,168]. Meanwhile, graphene-derived nanomaterials namely graphene oxide (GO) and reduced graphene oxide (rGO) are an excellent choice for electrode surface modification because they possess good mechanical, chemical, electronic, and thermal characteristics. Sengupta and Hussain [169] had extensively reviewed the application of graphene-based electrochemical biosensors for rapid detection of SARS-CoV-2 and other pathogenic viruses. 

Along with utilizing nanomaterials as the substrates for biorecognition element immobilization, the introduction of miniaturization technology makes the electrochemical biosensors suit for on-site diagnostics applications [160,170]. Due to the high demands for POCTs, miniaturized electrochemical sensors are broadly used to detect different target analytes including metal ions, small organic molecules, and biomolecules in diverse sectors such as food safety, environmental and healthcare [171]. Such sensor systems offer many benefits to electrochemical sensing platforms such as portability, simple sample preparation, ease to operate, good selectivity and sensitivity, rapid detection time, and low experimental cost [172,173]. Owing to their remarkable benefits, various designs of miniaturized electrochemical biosensors have been developed to detect SARS-CoV-2.

Last year, Alafeef et al. [108] research group developed the miniaturized electrochemical DNA sensor for quantitative measurement of SARS-CoV-2 in clinical samples. The electrochemical DNA sensor was fabricated using highly specific antisense thiolated ssDNA capped with gold nanoparticles (AuNPs) which target the viral nucleocapsid phosphoprotein (N gene) (Figure 8). The developed sensor can detect the SARS-CoV-2 RNA for only 5 min with a LOD of 6.9 copies/μL and a sensitivity of 231 copies/μL without any additional amplification technique. The obtained LOD in this study is comparable with other clinical approaches. Interestingly, the high sensitivity and specificity of this sensor manage to discriminate the positive COVID-19 samples from the negative ones with an accuracy of nearly 100%. Tripathy and Singh [66] proposed a miniaturized electrochemical sensor based on the DNA hybridization method for detecting SARS-CoV-2 as they successfully developed the label-free electrochemical DNA sensor for detection of dengue fever and breast/ovarian cancer in previous research [174,175]. The miniaturized device was fabricated on oxidized silicon substrates, using standard CMOS fabrication process flow and electrodeposition techniques. Such sensors utilized gold nanoparticles (AuNP) as the transducing element. In this sensing approach, a thiolated ssDNA probe which is complementary to SARS-CoV-2 RNA or its corresponding cDNA was immobilized on the surface of AuNP via gold-thiol self-assembly. Although this label-free miniaturized sensor could potentially become a POC biosensing device, additional steps such as the extraction of target DNA/RNA from the infected host and the following sample preparations are needed before the electrochemical detection. Such tedious sample preparation steps could affect the diagnostic accuracy and prolong the time required for analysis.

A further advance in the electrochemical sensing platform allows Mahari et al. [89] to develop an in-house built single printed carbon electrode (SPCE) called eCovSens. They fabricated the sensor by drop-casting the gold nanoparticles (AuNPs) onto the surface of the fluorine-doped tin oxide electrode (FTO) electrode before being allowed for immobilization with antibody probe. The addition of AuNPs on the surface of the FTO electrode will amplify the electrical signals of the fabricated sensor. Then, an nCOVID-19 monoclonal antibody probe was immobilized onto FTO/AuNPs electrode to recognize the spike antigen of nCOVID-19. This surface modification and antibody immobilization strategies provide rapid detection (10–30 s) with LOD of 90 fM using saliva samples. Furthermore, the eCovSens is an ultrasensitive sensor as they require only 20 μL sample volume compared to the other electrochemical biosensors that require at least 100 μL sample volume. Therefore, eCovSens could be a promising POC device as it is portable and stable until one month of storage.

Miripour et al. [116] presented a portable disposable sensor with an automatic electrochemical readout for real-time monitoring the reactive oxygen species (ROS) levels in the sputum of COVID 19 patients. Multi-Wall Carbon Nanotubes (MWCNTs) were selected as a nanomaterial to functionalized on the surface of the working electrode because they have good mechanical and electrical characteristics [176]. In this sensing system, the reaction between ROS molecules and the surface of the MWCNTs electrode generates an electrical signal measured via cyclic voltammetry. Such electrical signals correlate with the viral load in the sputum of COVID 19 samples. The responses of this disposable sensor were comparable with the CT scan results of the COVID-19 patient. In addition, the electrochemical ROS sensor could potentially use for rapid screening of patients that require prompt medical examination since it can give results less than 30 s. 

In recent work, Hashemi et al. [107] established the electrochemical nanosensor for screening SARS-CoV-2 glycoprotein in biological samples. They fabricated the nanosensor using a layer of GO coated with gold nanostars (AuNS). The addition of these excellent biochemical sensing compounds on the surface of the working electrode allows this nanosensor to demonstrate a good LOD (1.68 × 10^−22^ μg/mL) and sensitivity (0.0048 μAμg/mL/cm). More importantly, the developed nanosensor can detect the trace of viruses in less than 1 min without the requirement of any biological marker and sample pre-treatment. In addition, the fabricated nanosensor exhibits high sensitivity (95%) for the detection of unknown clinical SARS-CoV-2 samples. Such good performances promote the developed nanosensor as a rapid diagnostic test for the detection of viral disease.

Besides electrochemical sensing strategies, data transmission is another crucial aspect that needs to be ascertained for rapid deliverable results to final-user and at-home diagnosis [177]. Following the previous miniaturized sensing strategies, Chandra [178] had proposed the miniaturized label-free electrochemical sensor integrated with smartphones for rapid screening of SARS-CoV-2 (Figure 9). A smartphone-based “cloud” directory was suggested to provide real-time surveillance of this pathogenic virus through geo-tagging. Through this system, the sensors were capable of tracing the disease spreading around the world and enabling the form of a library of data and details required for future preparation to manage and control such pandemics. A similar strategy has been reported by Balaji et al. [114], where they developed an electrochemical biosensor-based smartphone for assessing COVID-19 patients. Herein, the electrochemical signals generated from the detection of RNA COVID-19 were transferred to the portable smartphone for displaying the results. The obtained results are stored in a database using the Internet of Things (IoT). Through an IoT system, the database can be shared with the concerned department to monitor the patients to avoid contact with other people and health surveillance of the infected patients. Accordingly, the risk of disease spreading can be reduced and expedited the best therapy for patients. In another study, Zhao et al. [115] developed the super sandwich-type electrochemical biosensor that can be adapted to a smartphone to detect SARS-CoV-2 RNA. They implemented a ‘plug and play method’ to produce a portable sensor that can easily be accessible by users to assess the diagnosis results conveniently. Interestingly, the fabricated portable electrochemical smartphone does not require further nucleic acid amplification to detect SARS-CoV-2, making them more convenient as a POC device. Moreover, Torrente-Rodríguez et al. [118] introduced the advanced wireless telemedicine-based electrochemical platform, SARS-CoV-2 RapidPlex, to screen COVID-19. Such sensing systems possess multiplexed platforms that provide information on the level of immune response (IgG and IgM), viral infection rate (nucleocapsid protein), and severity of disease (C-reactive protein) [179,180]. The amperometric data from the multiplexed sensor are recorded by custom PCB-based wireless potentiostat. The data are wirelessly transferred to a user smartphone via Bluetooth. This sensor can be used for at-home diagnosis since it is simple to operate and can give ultrasensitive detection of SARS-CoV-2.

### 4.2. Microfluidic Chip

The integration of microfluidic technology with the miniaturized electrochemical sensing platform can further accelerate the performance of virus detection. This device is a so-called microfluidic chip or lab-on-chip (LoC). Microfluidic technology revolutionizes the method for sampling, sample separation, mixing, chemical reaction, and electrochemical detection in one device. Such technology is beneficial for real-time detection, enabling multiplexing, and assembling multiple microfluidic components [76,181]. The use of microfluidic technology in an electrochemical sensing platform can significantly minimize the volume of samples as it can process a small volume of fluids by using tiny channels with dimensions at the microscale, usually tens to hundreds of micrometers [182]. Such advanced features allow the microfluidic electrochemical devices to emerge as great alternatives to conventional diagnostic methods due to their huge benefits such as fast detection times, better process control, automation, portability, reduced waste generation, low experimental cost and superior detection limit and sensitivity [183,184]. In addition, the microfluidic electrochemical devices can offer a high quality of pathogenic viral assessment since their physical and chemical environments can be precisely controlled [185]. The advancement in microfluidic technologies allows the integration of smart solutions such as the Internet of Medical Things (IoMT), e-health, artificial intelligence (AI), and machine learning to develop innovative healthcare technologies [186,187]. The microfluidic devices can be fabricated using various material bases such as polytetrafluoroethylene (PTFE), polydimethylsiloxane (PDMS), polycarbonate, silicon, glass, quartz, polymethyl methacrylate (PMMA), paper, hydrogel, three-dimensional (3D) printing and thermoset materials that subsequently offer diversity in their development [76,181,188].

To date, the application of microfluidic devices for SARS-CoV-2 detection are extensively reported in the literature. For instance, a 3D-nanoprinted COVID-19 microfluidic chip (3DcC) was presented by Ali et al. [103] (Figure 10). The integration of the PDMS microfluidic channel with a 3D electrochemical sensor allows the utilization of a small volume of antibodies (30 µL) and antigen (20 µL) to detect SARS-CoV-2. Such features enable the 3DcC device to detect SARS-CoV-2 in seconds with LOD of 2.8 × 10^−15^ for spike S1 protein and 16.9 × 10^−15^ M for its receptor-binding-domain (RBD) using a portable smartphone-based potentiostat. Moreover, Li et al. [105] developed a microfluidic paper-based analytical device (μPADs) to detect the IgG antibodies of SARS-CoV-2 in serum samples. In this research, only 3 μL of spiked human samples and antibodies are needed for the detection process completed within 30 min. A similar approach was applied by Yakoh et al. [104] to monitor the level of IgG and IgM of SARS-CoV-2. They fabricated the microfluidic chips using paper because it is cheaper, has a natural abundance and can be discarded by incineration after use. This is highly preferable for infectious disease diagnosing as the test kits need to be disposed of after testing. The electrochemical paper-based device, COVID-19 ePAD, can detect the targeted antibodies in serum of infected patients with 90% specificity and 100% sensitivity within 30 min. 

## 5. Conclusions

The existence of the COVID-19 pandemic encourages many researchers around the world to develop robust and accurate diagnostic tests for screening of SARS-CoV-2 to give immediate therapeutics and effective isolation of infected individuals. Up to now, the primary clinical diagnostic tests for COVID-19 have relied on the detection of viral nucleic acids, antigens, and antibodies specific for SARS-CoV-2. Due to the limitations of currently available diagnostics kits, electrochemical biosensors may be an excellent diagnostic tool for screening SARS-CoV-2 infection, as they meet the ASSURED criteria. The present review discusses current electrochemical sensing platform strategies for detecting SARS-CoV-2, including the use of the most sensitive biorecognition molecules, electrode surface modification with nanomaterials in conjunction with the miniaturized electrode and microfluidic technologies that could provide high-quality SARS-CoV-2 assessment. The majority of studies on electrochemical biosensors have used antibodies rather than nucleic acid as the biorecognition molecule since this method allows for easier sample preparation while maintaining good selectivity and sensitivity results. The antibodies were mainly immobilized on the surface of the working electrode, which had been modified using graphene-derived nanomaterials such as GO and rGo. The developed electrochemical biosensors can achieve the LOD as low as 2 copies/mL in clinical samples in less than 1 min of detection time without sample pre-treatment or labelling with any biorecognition molecule. The integration of the most sensitive electrochemical sensing platforms with smartphones enables the end-user to receive results quickly. The development of robust electrochemical biosensors for SARS-CoV-2 detection could be used to combat future pandemics. Although electrochemical biosensors have been proven for diagnosing a variety of SARS-CoV-2 clinical samples, they face a few limitations when moving from laboratory to on-site detection. Numerous issues, including the stability and reproducibility of the electrochemical biosensors, sample preparation processes, device engineering scale-up, and commercialization of electrochemical biosensors as a diagnostic device need to be further investigated. This aims to facilitate the electrochemical biosensors as a precise platform for rapid, simple, robust, and portable sensors for COVID-19 infection mass screening.

## Figures and Tables

**Figure 1 biosensors-12-00473-f001:**
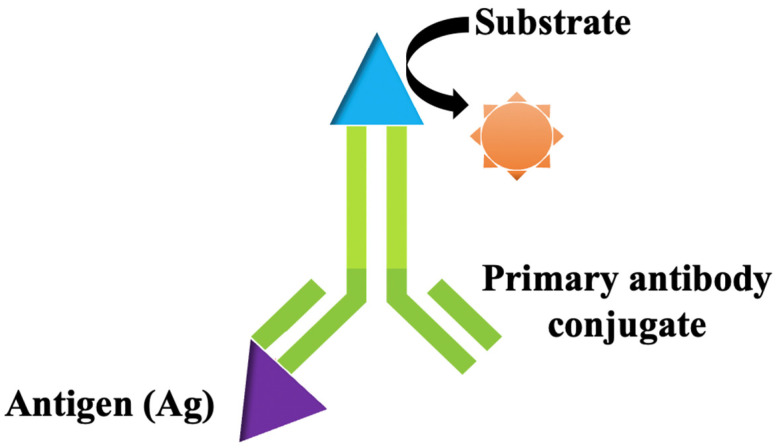
Schematic illustration of direct enzyme-linked immunosorbent assay (ELISA) that generates a color signal when an antibody binds to a specific antigen (protein).

**Figure 2 biosensors-12-00473-f002:**
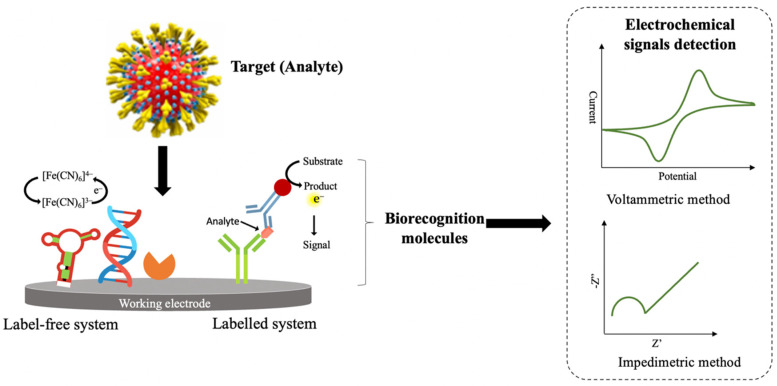
Schematic illustration of electrochemical biosensors platform based on label-free and labelled systems with various types of biorecognition molecules and electrochemical transducing techniques for the detection of SARS-CoV-2 in clinical samples. Adapted with permission from ref. [79]. Copyright 2020 Elsevier.

**Figure 3 biosensors-12-00473-f003:**
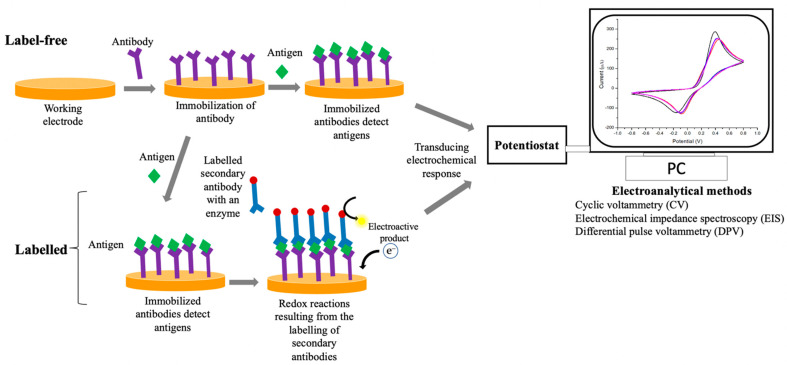
The schematic illustration for general fabrication of electrochemical immunosensor based on label-free and labelled systems (e.g., sandwich-type immunosensor) using gold electrode substrates.

**Figure 4 biosensors-12-00473-f004:**
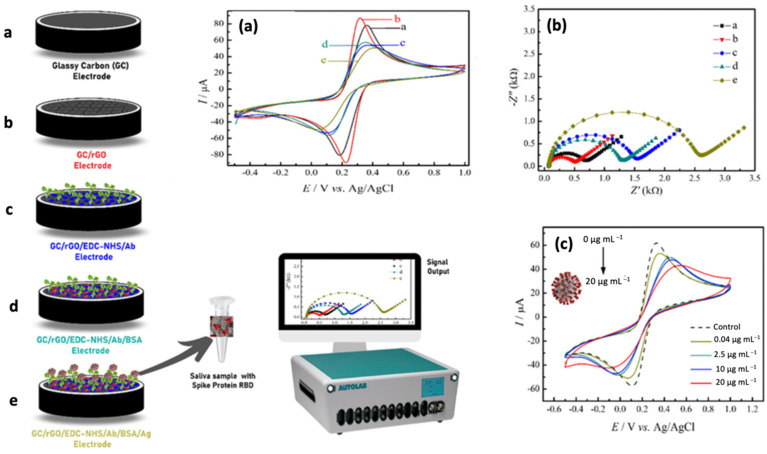
Schematic representation of fabrication steps for a label free impedimetric immunosensor for detection of SARS-CoV-2 in a saliva sample. (**a**) CV and (**b**) EIS measurements for each fabrication step in 0.2 mol/L PBS, pH 7.4, 0.1 mol/L KCl containing 5.0 mmol/L of [Fe(CN_6_)]^3−^/^4−^ for the working electrodes. (**c**) CV measurement of the immunosensor after the incubation with different antigen concentrations. Reproduced with permission from [97]. Copyright 2021 Multidisciplinary Digital Publishing Institute (MDPI).

**Figure 5 biosensors-12-00473-f005:**
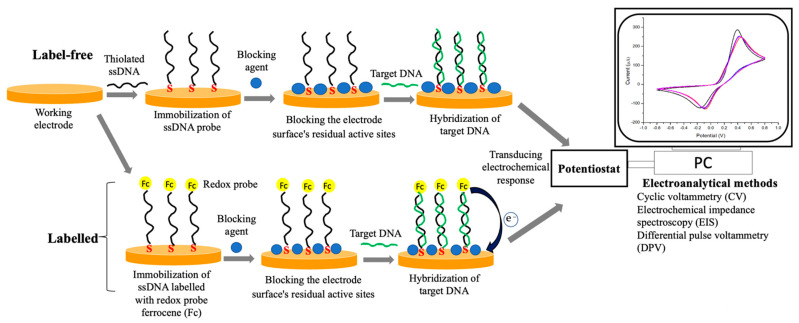
The schematic illustration for general fabrication of label-free and labelled electrochemical DNA sensors based on gold electrode substrates via self-assembly monolayer technique (thiol chemistry).

**Figure 6 biosensors-12-00473-f006:**
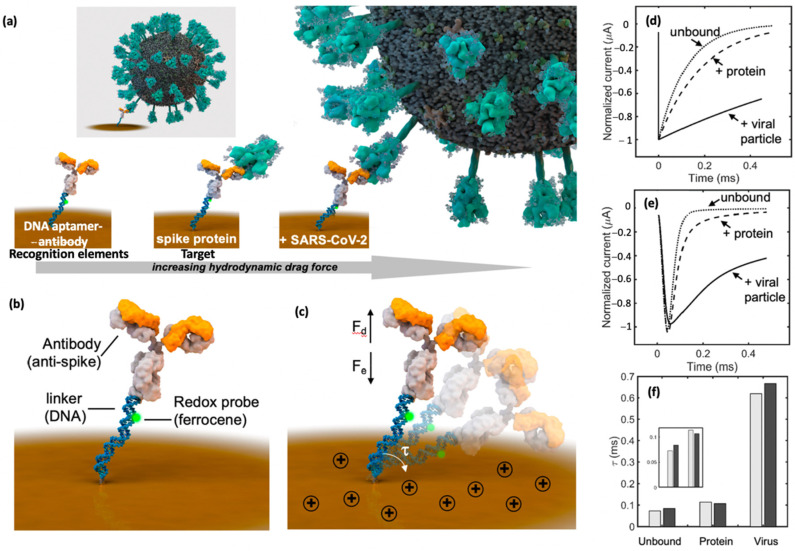
The fabrication of electrochemical sensors based on a labelled system using DNA aptamer-antibody conjugate as recognition elements for detection of SARS-CoV-2 virus. (**a**) Detection of SARS-CoV-2 viral particles by the fabricated sensor coated with gold on the electrode surface. (**b**) The design of the sensor consists of an analyte-specific antibody tethered to a linker composed of dsDNA that also includes the redox probe ferrocene. (**c**) The changes in electrical properties that occurred on the electrode sensor surface. (**d**–**f**) The peak chronoamperometric current of fabricated biosensor after exposure to target. The figure has been reproduced with permission from [100]. Copyright 2021 American Chemical Society (ACS).

**Figure 7 biosensors-12-00473-f007:**
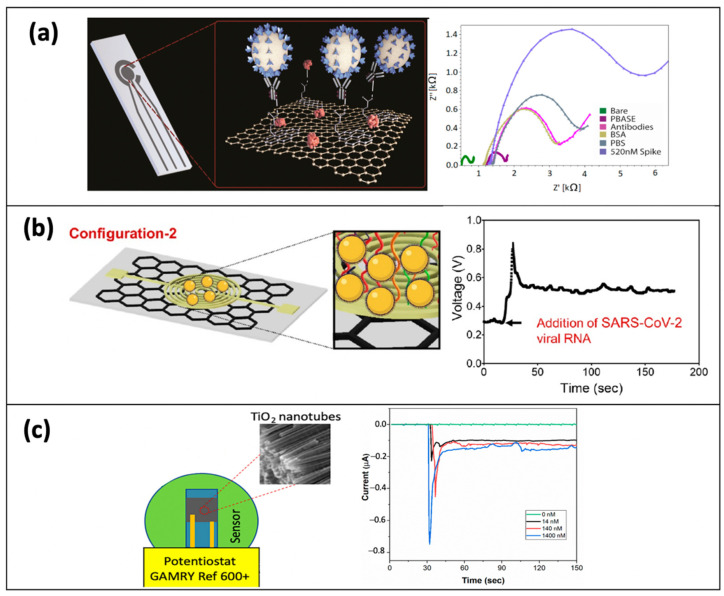
The surface modification of miniaturized electrochemical sensors with nanomaterials such as (**a**) graphene, (**b**) gold nanoparticle (AuNPs), and (**c**) cobalt-functionalized TiO_2_ nanotubes (Co-TNTs) together with its electrochemical measurements for rapid detection of SARS-CoV-2. (**a**) has been reproduced from [93] and (**c**) from [102] with permission from the Multidisciplinary Digital Publishing Institute (MDPI). (**b**) has been reproduced with permission from [108]. Copyright 2022 American Chemical Society (ACS).

**Figure 8 biosensors-12-00473-f008:**
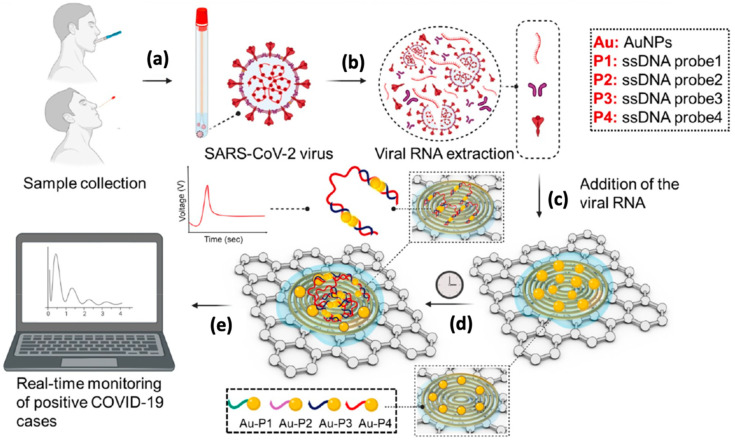
Schematic representation of the principal detection of label-free paper-based electrochemical DNA biosensors for SARS-CoV-2 detection in nasal swabs or saliva of the patients. (**a**) Step 1: Samples will be collected from the nasal swab or saliva of the infected individuals. (**b**) Step 2: The viral RNA of SARS-CoV-2 will be extracted from samples. (**c**) Step 3: The extracted RNA samples will be dropped onto the paper-based electrochemical DNA biosensor and (**d**) incubated for 5 min. (**e**) Step 4: The electrochemical measurement will be performed using a potentiostat. The figure has been reproduced with permission from [108]. Copyright 2022 American Chemical Society (ACS).

**Figure 9 biosensors-12-00473-f009:**
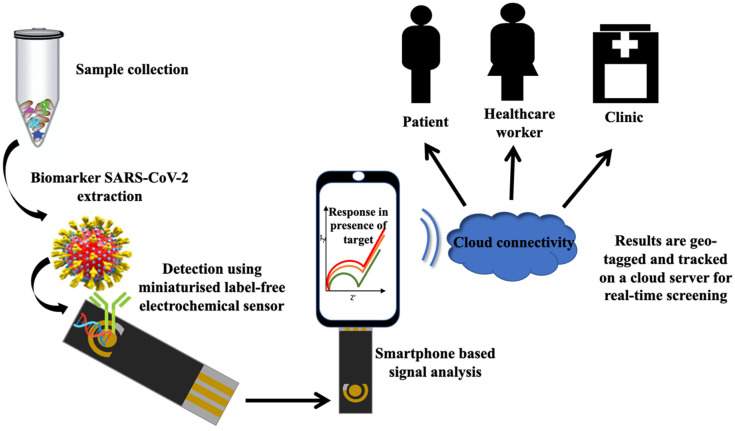
The miniaturized label-free electrochemical sensor is integrated with smartphone-based “cloud” directory for the real-time surveillance of COVID-19 through geo-tagging. This figure has been adapted with permission from ref. [178]. Copyright 2020 Elsevier.

**Figure 10 biosensors-12-00473-f010:**
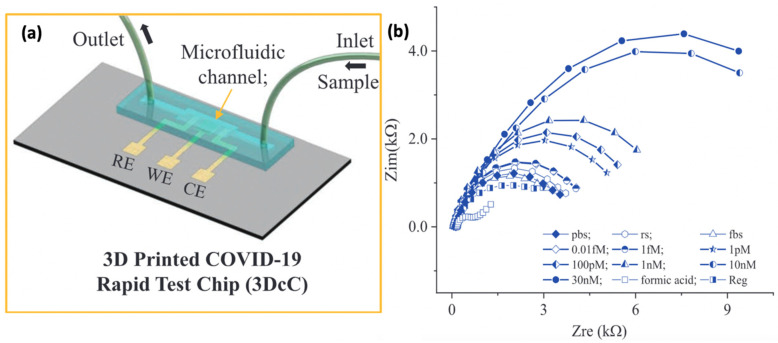
(**a**) A 3D-nanoprinted COVID-19 microfluidic chip (3DcC) that fabricated using PDMS. (**b**) The detection of 3DcC at different concentrations of SARS-CoV-2 antibodies in PBS solution using the electrical impedance spectroscopy (EIS) method. This figure has been reproduced with permission from [103]. Copyright 2022 John Wiley and Sons.

**Table 2 biosensors-12-00473-t002:** Several reported electrochemical biosensors for SARS-CoV-2.

Target Analyte	Recognition Element	Electrode Modification	Platform Technology	Name	Sample Type	Integration with Smartphone	Electrochemical Detection Method	Response Time	Limit of Detection	Reference
Spike protein	Monoclonal antibody	Fluorine doped tin oxide electrode with gold nanoparticles	Screen-printed carbon electrode	eCovSens	Saliva	No	DPV	10–30 s	90 fM	[89]
	Antibody	Laser-scribed graphene electrode combined with three-dimensional gold nanostructures	Miniaturise laser-scribed graphene electrode	-	Serum	Yes	DPV	1 h	2.9 ng/mL	[90]
	Antibody	Graphene	Graphene-field effect transistor	COVID-19 field effect transistor sensor	Nasopharyngeal	No	-	>1 min	1 fg/mL in antigen protein 1.6 × 10^1^ pfu/mL in culture medium 2.42 × 10^2^ copies/mL in clinical samples	[91]
	IgG antibody	Cu_2_O nanocubes Staphylococcal protein A	Screen-printed carbon electrode	Nanobiodevice	Saliva and artificial nasopharyngeal	No	CV, EIS	20 min	0.04 fg/mL	[92]
	Monoclonal antibody	Graphene	Screen printed electrode	-	-	No	CV, EIS	45 min	260 nM	[93]
	Human angiotensin-converting enzyme	Gold nanoparticles	Graphite printed electrode	Low-cost Electrochemical Advanced Diagnostic (LEAD)	Saliva, nasopharyngeal swab	No	SWV	6.5 min	229 fg/mL	[94]
	DNA Aptamer	Gold electrode		Electrochemical-aptamer-based (EAB) sensor	Serum and artificial saliva	No	SWV	15 s	-	[95]
	Antibody	Glassy carbon electrode-gold cluster	-	-	Saliva and oropharyngeal swab	No	CV, SWV	~35 min	0.01 ag/mL	[96]
	Monoclonal antibody	Glassy carbon electrode-reduced graphene oxide	-	-	Saliva	No	CV, EIS, SWV	-	150 ng/mL	[97]
	Angiotensin-converting enzyme-2 (ACE2)	Carbon electrode-Nafion permeable membrane	Screen-printed electrode	RAPID 1.0	Saliva and Nasopharyngeal/oropharyngeal swab	Yes	CV, EIS	4 min	1.16 PFU/mL	[98]
	IgG antibody	Graphene electrode	Screen-printed electrode (cellulose paper substrate)		Nasopharyngeal swab	Yes	CV, EIS	-	0.25 fg/mL	[99]
Spike protein and viral particles	DNA-spike antibody conjugate	Electrode-tethered sensors	-	-	Saliva	No	Chronoamperometry (CA)	5 min	-	[100]
Spike protein and receptor-binding domain	Monoclonal antibody	ACEA Bioscience’s 96-well platform integrated with sensing electrode	-	Serum	No	EIS	<5 min	-	-	[101]
	-	Cobalt-functionalized titanium dioxide nanotubes	Custom-cobalt-titanium dioxide nanotubes packaged printed circuit board setup	-	-	No	Amperometry	30 s	~0.7 nM	[102]
	Antibodies	3D nanoprinting of electrodes coated by reduced-graphene oxide	Microfluidic chip	3D-printed COVID-19 test chip (3DcC)	-	Yes	EIS	~ 11.5 s	2.8 fM for S protein 16.9 fM for RBD	[103]
	Antibodies	Graphene oxide	Folding paper-based electrochemical sensor	COVID-19 ePAD	Serum	Yes	SWV	30 min	0.11 ng/mL	[104]
	IgG antibody	Zinc oxide nanowires	Microfluidic paper-based analytical devices (μPADs)	-	Serum	No	EIS	15 min	-	[105]
	ssDNA aptamer	Screen-printed carbon electrodes-gold nanoparticles	Screen-printed electrode	-	-	No	EIS	40 min	66 pg/mL	[106]
S1 and S2 glycoproteins	-	Graphene oxide and gold nanostars	Screen-printed electrode	-	Blood, saliva and nasopharyngeal swab	No	CV, DPV	1 min	1.68 × 10^−22^ µg/mL	[107]
Nucleocapsid phosphoprotein	ssDNA	Gold nanoparticle and graphene nanoplatelets	Paper-based electrochemical platform	-	Nasopharyngeal and saliva	No	CV	<5 min	6.9 copies/μL	[108]
	Antibody	Carbon nanofiber	Screen-printed carbon electrode coating with absorbing cotton padding	Cotton-tipped electrochemical immunosensor	Nasopharyngeal swab	Yes	SWV	~20 min	0.8 pg/mL	[109]
	ssDNA	Indium doped tin oxide-polypyrrole-gold nanoparticles	Screen-printed indium doped tin oxide electrode	-	Nasopharyngeal swab	No	CV, EIS	15 min	258.01 copies/µL	[110]
Nucleocapsid gene amplicons	-	Gold electrode	Printed circuit-board-based lab-on-chips	-	-	No	CV, DPV	-	10 pg/μL (approximately 1.7 fM	[111]
Nucleocapsid and spike protein	One-step sandwich hybridization of isothermal rolling circle amplification amplicons	-	Screen-printed carbon electrode	-	Nasopharyngeal swab sample	No	DPV	30 min <2 h from RNA extraction to the detection step	1 copy/µL of N and S gene	[112]
	Antibody	Magnetic bead-based immunosensor combined with carbon black nanomaterial	Screen-printed electrode	-	Saliva	No	DPV	30 min	19 ng/mL for S protein 8 ng/mL for N protein	[113]
RNA	-	Gold	-	-	Nasopharyngeal	Yes	-	70-80 s	Accuracy of 81%	[114]
	Replicase complex (ORF1ab)	p-sulfocalix[8]arene functionalized graphene	Screen-printed carbon electrode	-	Throat swab, urine, feces, serum, saliva	Yes	DPV	<10 s	200 copies/mL	[115]
Reactive oxygen species	-	Multi-wall carbon nanotubes decorated electrode	Portable automatic electrochemical readout board and a sensing disposable sensor	COVID-19 associated ROS diagnosis (CRD)	Sputum	No	CV	<30 s	Accuracy: 97% Sensitivity: 97%	[116]
Recombinant protein with anti-GFP nanobody	Nanobodies	Gold organic transistors	Nanobody-organic electrochemical transistors (OECT) disposable platform	-	Nasopharyngeal swab and saliva	No	CV, EIS	10 min <15 min from sample to result	1.2 × 10^−21^ M in saliva 1.8 × 10^−20^ M in buffer	[117]
Antigen nucleocapsid protein, IgM and IgG antibodies, inflammatory biomarker C-reactive protein	Capture antigens and antibodies	Laser-engraved graphene	Multiplexed telemedicine platform system with a graphene sensor array connected to a printed circuit board for signal processing and wireless communication	SARS-CoV-2 RapidPlex	Serum and saliva	Yes	DPV, open-circuit potential-electrochemical impedance spectroscopy (OCP-EIS)	~1 min	-	[118]
ORF1ab fragment	Catalytic hairpin assembly and terminal deoxynucleotidyl transferase mediated-DNA polymerization	Gold electrode	-	-	Serum and saliva	No	EIS, DPV	-	26 fM	[119]

CV-Cyclic voltammetry; DPV-Differential pulse voltammetry; SWV-Square wave voltammetry; EIS-Electrochemical impedance spectroscopy.

**Table 3 biosensors-12-00473-t003:** Advantages and drawbacks of common biorecognition elements applied in the fabrication of electrochemical biosensors for SARS-CoV-2 detection.

Type of Electrochemical Biosensors	Biorecognition Elements	Binding Interaction	Advantages	Drawbacks
Nucleic acid-based	ssDNA/RNA	DNA-DNA, DNA-RNA	Detection of ssDNA PCR products, simple to produce, stable, very specific, ability to miniaturize, easy to implement	Restricted for gene sequence detection, strict to hybridization conditions and expensive
	Aptamer	Aptamer-binding protein Aptamer-DNA Aptamer-antibody	Small size, low-cost, stable, simple to produce, high affinity and selectivity, wide variety of targets	Strict to hybridization conditions, long-term SELEX process and may require additional complex steps
Immunosensor	Monoclonal antibodies (mAb)	Non-covalent interaction between antibody-antigen/protein	More specific than pAb, low chances of cross-reaction	High cost, unstable (very sensitive to environmental conditions) and complex production
	Polyclonal antibodies (pAb)		Low production cost, various epitopes and mass-produce	Unstable (very sensitive to environmental conditions) and high chances of cross-reaction
	Antibody single chain Fv fragments (scFv)		Small size compared with the whole antibody and low variability	Longer time to produce, lower affinities compared with whole antibodies and not applicable for small molecules

Adapted from reference [74,79].

## Data Availability

Not applicable.

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
