# Peer review of "Utilizing Electrochemical-Based Sensing Approaches for the Detection of SARS-CoV-2 in Clinical Samples: A Review"

_biosensors, 2022, doi:10.3390/bios12070473_

Round 1
Reviewer 1 Report
I have carefully read the manuscript entitled Utilizing electrochemical based sensing approaches for the detection of SARS-CoV-2 in clinical samples: A review.
Overall, the reivewer suggested the manuscipt should be suitable for the publications in Biosensors.
Author Response
Dear reviewer 1,
Thanks for review our paper and accepted the content.
Thanks
Reviewer 2 Report
The review by Zambry et al is very interesting as it highlights the current research on diagnosis of SARS-Cov2 using facile techniques of biosensors. Although the detection SARS-Cov2 via the use of biosensors is still in its infancy stages, the current review thus shows the a potential of these well-known devices to contribute in the fight against the spread of the virus.
As such this reviewer recommends consideration for publication after minor revisions according to the following comments and criticisms;
General Comments:
1. Provide CT scans of the infected patient at various stages.
2. Schematic diagrams of ELISA test can provide the reader with the clue of the technique.
3. From line 189, is it ASSURED or ASSURANCE?
4. Lines 244-251 should be the opening paragraphs to section 3, then followed by lines 228-243.
5. Consistency in the use of chemical names and/or formulaes is recommended in the tables. Otherwise, chemical names can be provided in the footnote.
6. ITO is an acronym for Indium doped Tin Oxide, not the one shown in Table 2 for ref 120.
7. Since it is stated that CV determines the preliminary redox properties, what do the DPV and SWV indicate? Also, to what concentration levels can the CV be useful for the detection of the virus? Have DPV or SWV been used for SARS-Coc-2 or its sister viruses and what have been the detection limits and response times?
8. There is no mention of observations from refs in Table 2. Additionally, what were the observations of Abrego-Martinez et al. ?
9. Revision of lines 384- 397 (page16) is recommended as the paragraph is confusing.
10. For both Liv et al and Zacariotto et al, were the biosensors tested in real samples or were the results based on the biomakers?
11. From Ishikawa et al, what were the LoD and response time?
Author Response
Dear reviewer 2,
Thanks for review our paper and really appreciate. The feedback as file attachment.
Thanks

Reviewer 3 Report
The manuscript “Utilizing electrochemical based sensing approaches for the detection of SARS-CoV-2 in clinical samples: A review” by Zambry et al. reviews the fundamentals of fabricating electrochemical biosensors, such as the chosen electrode materials, electrochemical transducing techniques, and sensitive biorecognition molecules. This work is well written and could be accepted for publication after minor revision. Here are the comments and suggestions:
1. In Fig. 5, receptors for label-free system should be labeled.
2. In Table 2, some targets and recognition elements are misplaced on pages 11 and 12. This table should be reorganized with the list of the same target or detection method.
3. Some typical results for CV, EIS and/or DPV can be showed on the display of the PC in Figs. 2 and 3.
4. “p”rotein is missing from Fig. 5(e).
5. Line 445, the concentration of LOD should be corrected.
Author Response
Dear reviewer 3,
Thanks for review our paper and really appreciate. The comment and feedback for the comment as file attachment.
Thanks

Reviewer 4 Report
Dear Sir,
The article entitled "Utilizing electrochemical based sensing approaches for the detection of SARS-CoV-2 in clinical samples: A review” by the authors Nor Syafirah Zambry et al. is quite interesting for the scientific community especially for researchers involved in sensors for detecting Covid 19. The paper describes all the electrochemical sensors used for the detection of SARS-CoV-2 and also describes the promising future of miniaturized electrochemical biosensors integrated with microfluidics for detection of virus.
The article is ready to be accepted in present form.
Author Response
Dear reviewer 4,
Thanks for accept our paper and really appreciate.
Thanks